# A miRNA-based diagnostic model predicts resectable lung cancer in humans with high accuracy

Keisuke Asakura[1,2,10], Tsukasa Kadota [3,4,10], Juntaro Matsuzaki [3], Yukihiro Yoshida [1], Yusuke Yamamoto[3], Kazuo Nakagawa[1], Satoko Takizawa[3,5], Yoshiaki Aoki[6], Eiji Nakamura[6], Junichiro Miura[6], Hiromi Sakamoto[7], Ken Kato [8], Shun-ichi Watanabe[1] & Takahiro Ochiya[3,9 ✉]

Lung cancer, the leading cause of cancer death worldwide, is most frequently detected through imaging tests. In this study, we investigated serum microRNAs (miRNAs) as a possible early screening tool for resectable lung cancer. First, we used serum samples from participants with and without lung cancer to comprehensively create 2588 miRNAs profiles; next, we established a diagnostic model based on the combined expression levels of two miRNAs (miR-1268b and miR-6075) in the discovery set (208 lung cancer patients and 208 non-cancer participants). The model displayed a sensitivity of 99% and specificity of 99% in the validation set (1358 patients and 1970 non-cancer participants) and exhibited high sensitivity regardless of histological type and pathological TNM stage of the cancer. Moreover, the diagnostic index markedly decreased after lung cancer resection. Thus, the model we developed has the potential to markedly improve screening for resectable lung cancer.

[1] Department of Thoracic Surgery, National Cancer Center Hospital, 5-1-1 Tsukiji, Chuo-ku, Tokyo 104-0045, Japan. [2] Division of Thoracic Surgery, Department of Surgery, Keio University School of Medicine, 35 Shinanomachi, Shinjuku-ku, Tokyo 160-8582, Japan. [3] Division of Molecular and Cellular Medicine, National Cancer Center Research Institute, 5-1-1 Tsukiji, Chuo-ku, Tokyo 104-0045, Japan. [4] Division of Respiratory Diseases, Department of Internal Medicine, The Jikei University School of Medicine, 3-25-8 Nishi-shimbashi, Minato-ku, Tokyo 105-8461, Japan. [5] Toray Industries, Inc. 6-10-1 Tebiro, Kamakura city, Kanagawa 248-0036, Japan. [6] Dynacom Co., Ltd., World Business Garden E25, 2-6-1 Nakase, Mihama-ku, Chiba city, Chiba 261-7125, Japan. [7] Department of Biobank and Tissue Resources, National Cancer Center Research Institute, Tokyo 104-0045, Japan. [8] Department of Gastrointestinal Medical Oncology, National Cancer Center Hospital, 5-1-1 Tsukiji, Chuo-ku, Tokyo 104-0045, Japan. [9] Institute of Medical Science, Tokyo Medical University, Tokyo 160-0023, Japan. [10] These authors contributed equally: Keisuke Asakura, Tsukasa Kadota. ✉email: tochiya@ncc.go.jp

Lung cancer is the most common cancer and the leading cause of cancer-related death worldwide[1]. The 5-year overall survival rate in patients with stage I non-small cell lung cancer is approximately 80%[2]. However, most patients with stage I lung cancer are asymptomatic and are thus unlikely to be diagnosed. In fact, about 75% of lung cancers are diagnosed as locally advanced or metastatic disease (stage III or IV), which are associated with poorer overall survival (37 and 6%, respectively)[2]. Therefore, early detection is essential for decreasing lung cancer-related mortality.

Computed tomography (CT) has been shown to be an effective method for lung cancer screening of high-risk populations. The National Lung Screening Trial revealed that high-risk participants who underwent CT screening had a 20% decrease in lung cancer mortality[3]. Based on this result, in 2014, the United States Preventive Services Task Force (USPSTF) released recommendations for low-dose CT scans for lung cancer screening in high-risk patients (people 55–74 years old with a 30 pack-year smoking history who currently smoke or quit within the past 15 years)[4]. However, screening by CT scan has several limitations: low specificity (61%) for detection of lung cancer, resulting in unnecessary follow-up CT scans or invasive lung biopsies[5]; poor access to CT facilities in some regions[6]; and a low screening rate (3.9% in the eligible high-risk population in the United States)[7].

MicroRNAs (miRNAs), 19–22-nucleotide noncoding RNAs that regulate gene activity, are differentially regulated in various types of cancer, including ovarian, liver, gastric, pancreatic, esophageal, colorectal, breast, and lung cancers[8,9]. Due to the marked stability of miRNA, it is measurable in whole blood[10], plasma[11–13], serum[14,15], and sputum[16,17], and is therefore suitable as a potential biomarker for lung cancer. Previous reports revealed the usefulness of circulating miRNAs for detection of lung cancer[8,13–15,18]. However, the results of these studies were discrepant, possibly due to the limited number of samples or analyzed miRNAs. To address this issue, we managed a national research project in Japan called "Development and Diagnostic Technology for Detection of miRNA in Body Fluids." The purposes of this project were to standardize platforms for the analysis of serum miRNAs and to characterize the serum miRNA profiles in 13 types of cancer, including lung cancer, using a large sample size ($n > 50,000$). The project was the largest study to date of miRNA in the context of cancer screening. In the study reported here, we examined the utility of serum miRNAs as biomarkers for the detection of resectable lung cancer using a large number of serum samples.

## Results

**Participants**. To generate comprehensive miRNA expression profiles, 1698 lung cancer and 207 non-cancer serum samples from National Cancer Center (NCC) Biobank and 1998 non-cancer serum samples from Yokohama Minoru Clinic (YMC) were analyzed by miRNA microarray. Among the 1698 lung cancer serum samples, 74 were excluded due to low-quality microarray results, 33 due to past history of other cancers, 25 due to lack of patient information, 19 due to treatment before collection of serum, and 4 because the interval between serum collection and surgery was greater than 180 days, leaving 1566 samples for analysis (Fig. 1).

Lung cancer, NCC non-cancer, and YMC non-cancer samples were grouped into discovery and validation sets. The discovery set included 208 lung cancer, 104 NCC non-cancer, and 104 YMC non-cancer samples. The validation sets included 1358 lung cancer, 103 NCC non-cancer, and 1867 YMC non-cancer samples. Patient characteristics for the discovery and validation sets are shown in Table 1. In the discovery set, we observed no significant difference in patient characteristics, including age, sex, or smoking history, between the 208 lung cancer patients and 208 non-cancer participants. In the validation set, age was significantly higher in the 1358 lung cancer patients than in the 1970 non-cancer participants ($66.3 \pm 0.3$ vs. $50.3 \pm 0.2$, $p < 0.001$). In the validation set, the proportions of men and smokers were also significantly higher among lung cancer patients than among non-cancer participants (57.6% vs. 52.1%, $p = 0.002$; 65.4% vs. 20.3%, $p < 0.0001$, respectively).

**Selection of circulating miRNA biomarker candidates**. The expression levels of the 2588 miRNAs were measured in the discovery set (208 lung cancer and 208 non-cancer samples). Of those, 406 miRNAs passed the quality check criteria, described in

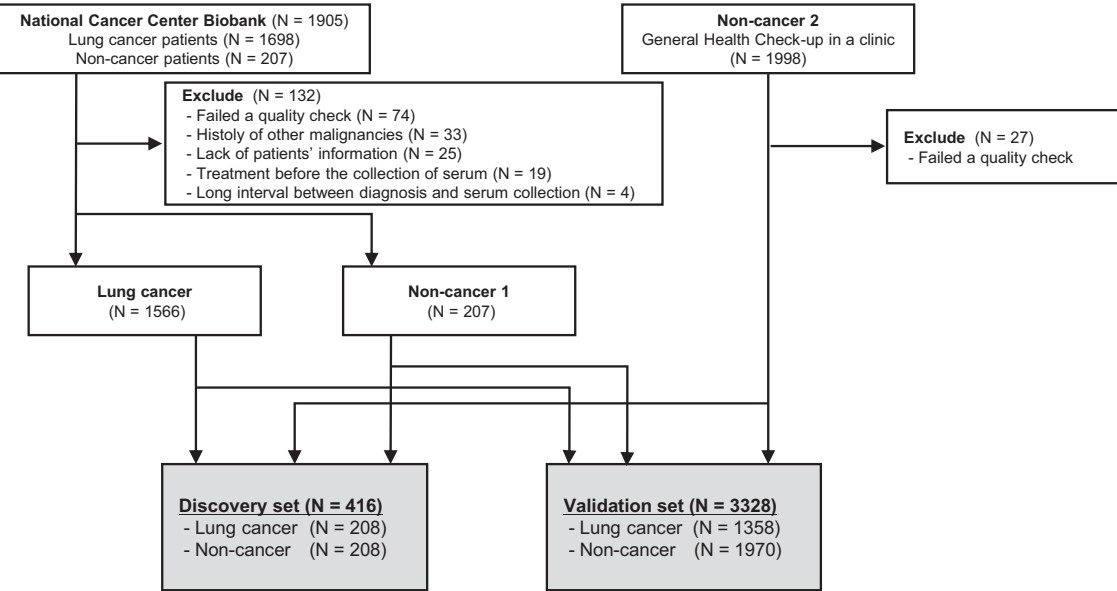

**Fig. 1 Workflow of lung cancer and non-cancer controls used for developing a diagnostic model.** Serum samples were obtained from 1698 lung cancer, 207 non-cancer patients in National Cancer Center biobank, and 1998 non-cancer participants in Yokohama Minoru Clinic. After excluding 132 samples from National Cancer Center biobank and 27 samples from Yokohama Minoru Clinic, the sample set was divided into two groups: the discovery set ($N = 416$) and validation set ($N = 3328$).

**Table 1 Patient characteristics in discovery and validation sets.**

| Variables | Discovery set (*N* = 416) | | | Validation set (*N* = 3328) | | |
|---|---|---|---|---|---|---|
| | Lung cancer (*N* = 208) | Non-cancer (*N* = 208) | *P* Value | Lung cancer (*N* = 1358) | Non-cancer (*N* = 1970) | *P* Value |
| Age, years ± SD | 60.2 ± 0.5 | 60.0 ± 0.5 | 0.7843 | 66.3 ± 0.3 | 50.3 ± 0.2 | <0.001 |
| Sex, *n* (%) | | | | | | |
| Men | 113 (54.3) | 103 (50.0) | 0.377 | 782 (57.6) | 1026 (52.1) | 0.002 |
| Women | 95 (45.7) | 105 (50.0) | | 576 (42.4) | 944 (47.9) | |
| Smoking status, *n* (%) | | | | | | |
| Former or current | 84 (40.4) | 83 (40.0) | 1.0 | 888 (65.4) | 399 (20.3) | <0.001 |
| Never | 124 (59.6) | 125 (60.1) | | 470 (34.6) | 1571 (79.7) | |
| Histology, n (%) | | | | | | |
| Adenocarcinoma | 179 (86.1) | – | – | 1038 (76.4) | – | – |
| Squamous carcinoma | 14 (6.7) | – | – | 207 (15.2) | – | – |
| Adenosquamous carcinoma | 1 (0.5) | – | – | 17 (1.3) | – | – |
| Small cell carcinoma | 1 (0.5) | – | – | 22 (1.6) | – | – |
| Other | 13 (6.3) | – | – | 74 (5.4) | – | – |
| Stage, n (%) | | | | | | |
| I (IA+ IB) | 155 (74.5) | – | – | 971 (71.5) | – | – |
| II (IIA+ IIB) | 26 (12.5) | – | – | 207 (15.2) | – | – |
| III (IIIA+ IIIB) | 22 (10.6) | – | – | 170 (12.5) | – | – |
| IV | 3 (1.4) | – | – | 8 (0.6) | – | – |
| 0 | 2 (1.0) | – | – | 2 (0.1) | – | – |

methods, and were selected for subsequent analysis (Fig. 2a). Principal component analysis mapping (Fig. 2b) and hierarchical unsupervised clustering analysis (Fig. 2c) were performed to visualize the expression patterns of these 406 miRNAs in all samples in the discovery set. These two classification analyses showed that the miRNA profiles differed between lung cancer and non-cancer samples.

**Identifying the best combination of miRNAs for lung cancer detection**. To establish a comprehensive discriminant consisting of one to three miRNAs in the discovery set, we used Fisher's linear discriminant analysis (Table 2, Supplementary Tables 1 and 2). One single miRNA was effective in distinguishing cancer patients (diagnostic index = 0.491213 × miR-17-3p – 2.49845; Area under the curve (AUC), 0.935; sensitivity, 93.3%; specificity, 88.5%) (Fig. 3a). However, a combination of two miRNAs (miR-1268b and miR-6075) improved the AUC relative to the single miRNA (diagnostic index = −3.56049 × miR-1268b + 1.99039 × miR-6075 + 16.7999; AUC, 0.993; sensitivity, 99.0%; specificity, 99.0%) (Fig. 3b, c). The addition of the first candidate miRNA (miR-6858-5p) to the combination of two miRNAs (miR-1268b and miR-6075) did not improve the results (AUC, 0.995; sensitivity, 99.0%; specificity, 99.0%) (Table 2). Therefore, we selected the combination of two miRNAs (miR-1268b and miR-6075) that yielded the best discrimination in the discovery set.

We then confirmed the diagnostic performance of the model in the validation set, which showed that the model was highly reliable (AUC, 0.996; sensitivity, 95.0%; specificity, 99.0%; Fig. 4). According to our univariable logistic regression analysis, the odds ratio (OR) of the diagnostic model for the presence of lung cancer was 21.76 (95% confidence interval [CI], 15.98–29.63). Because patient sex, age, and smoking status were not matched between cases and controls, we performed sex-, age-, and smoking status–adjusted logistic regression analysis of the validation set. The results revealed that the diagnostic indices of the two miRNAs were statistically significantly associated with the presence of lung cancer (adjusted OR, 20.34; 95% CI, 13.99–29.57) (Supplementary Table 3).

**Performance of the diagnostic index as a function of clinical condition**. Next, we examined the performance of the diagnostic index for each pathological TNM stage and each histological type of lung cancer in the validation set. The diagnostic index exhibited high performance for all pathological stages (IA, 96.1%; IB, 93.7%; IIA, 97.3%; IIB, 96.7%; IIIA, 90.2%; IIIB, 83.3%; IV, 100%), T stages (T1a, 96.1%; T1b, 95.6%; T2a, 93.6%; T2b, 92.3%; T3, 94.4%; T4, 94.1%), N stages (N0, 95.5%; N1, 95.8%; N2, 90.1%), M stages (M0, 94.7%; M1a, 100%), and histological types (adenocarcinoma, 95.1%; squamous cell carcinoma, 94.2%; small-cell lung cancer, 90.9%) (Fig. 5).

**Comparison of the diagnostic indexes between preoperative and postoperative serum samples**. Next, we compared the diagnostic indexes of three miRNAs between preoperative and postoperative serum samples from 180 lung cancer patients. The diagnostic indexes of miR-17-3p, miR-1268b, and miR-6075 and the two-miRNA panel (miR-1268b and miR-6075) were significantly decreased after surgery. (miR-17-3p, 0.71 ± 0.46 vs. −3.10 ± 0.32, $p < 0.001$; miR-1268b, 1.15 ± 0.88 vs. −1.83 ± 0.75, $p < 0.001$; miR-6075, 0.44 ± 0.85 vs. −2.10 ± 1.16, $p < 0.001$; two-miRNA panel (miR-1268b and miR-6075), 0.98 ± 0.82 vs. −4.30 ± 1.15, $p < 0.001$; Fig. 6).

**Discussion**

In this study, we comprehensively profiled the expression of 2588 miRNAs, constituting all human miRNAs identified to date, according to miRBase release 21 (http://www.mirbase.org/)[19], in serum samples from 1566 lung cancer and 2178 non-cancer participants, on a standardized microarray platform. The results revealed that lung cancer patients could be accurately distinguished from non-cancer participants based on the serum levels of two miRNAs (sensitivity and specificity, 99%). This is the largest study to date of miRNA profiling in the context of lung cancer detection.

Recently, a series of articles reported that many kinds of circulating miRNAs can be applied to lung cancer detection[8,10,11,14,15,20,21].

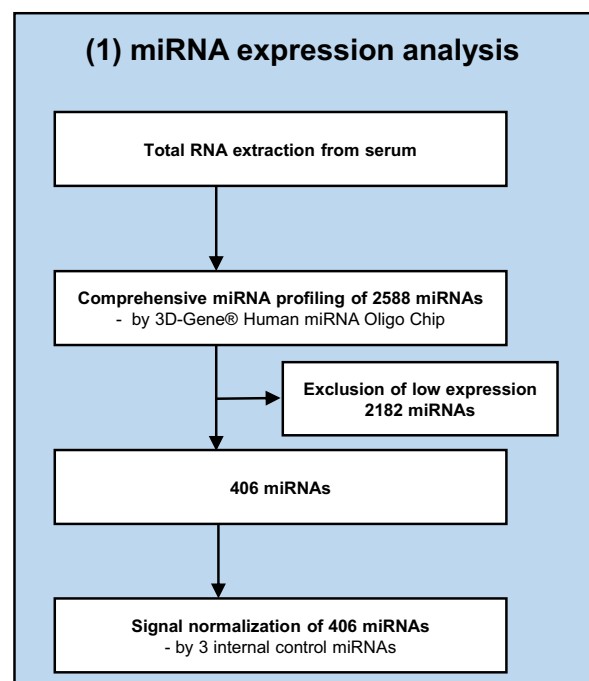
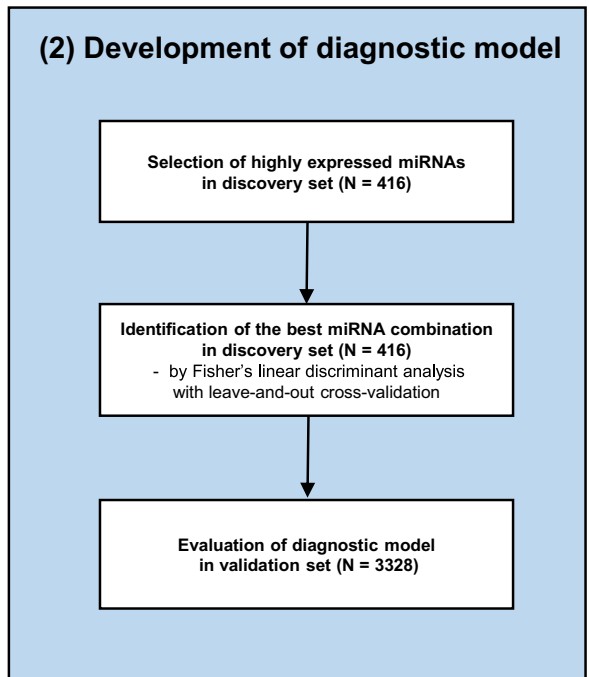

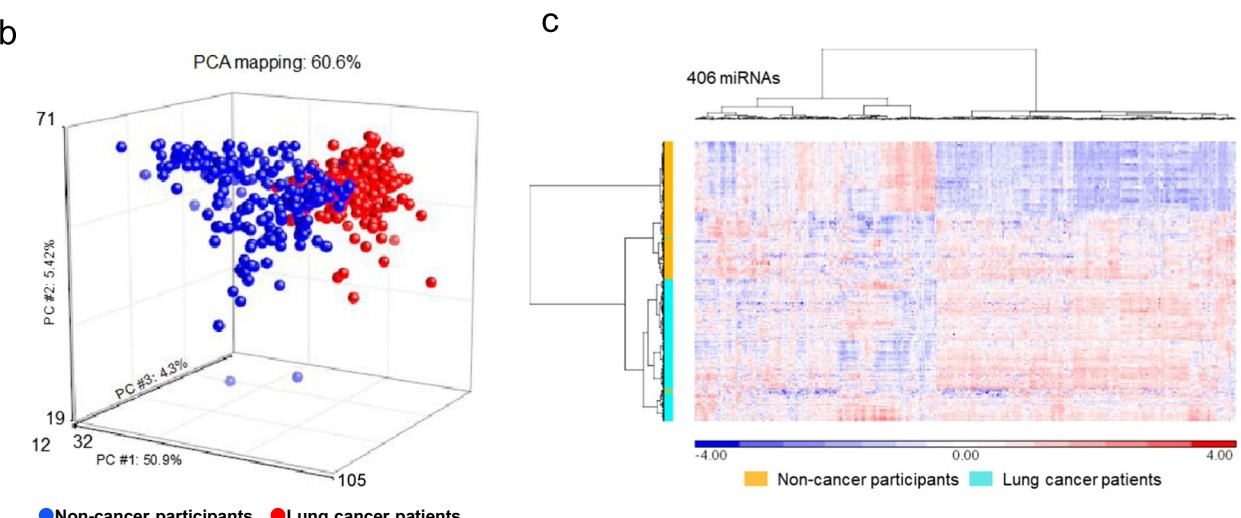

●**Non-cancer participants** ●**Lung cancer patients**

**Fig. 2 Strategy for the selection of candidate miRNAs for lung cancer diagnosis. a** Flow diagram of the (1) miRNA expression analysis and (2) development of the diagnostic model. **b** A principal component analysis map for 208 lung cancer samples and 208 non-cancer samples with 406 miRNAs. **c** Heat map showing the differences in miRNA expression levels between 208 lung cancer and 208 non-cancer control samples.

**Table 2 Discriminant analysis for lung cancer (diagnostic model).**

| Model | Number of miRNAs | Sensitivity (%) | Specificity (%) | Accuracy (%) | PPV (%) | NPV(%) | AUC |
|---|---|---|---|---|---|---|---|
| model 1 | 1 | 93.3 | 88.5 | 90.9 | 89.0 | 92.9 | 0.935[a] |
| model 2 | 2 | 99.0 | 99.0 | 99.0 | 99.0 | 99.0 | 0.993[b] |
| model 3 | 3 | 99.0 | 99.0 | 99.0 | 99.0 | 99.0 | 0.995 |

model1: $(0.491213) \times$ miR-17-3p $- 2.49845$.
model2: $(-3.56049) \times$ miR-1268b $+ (1.99039) \times$ miR-6075 $+ 16.7999$.
model3: $(-3.87517) \times$ miR-1268b $+ (2.28137) \times$ miR-6075 $+ (-0.81776) \times$ miR-6858-5p $+ 23.5615$.
[a]DeLong's test for two correlated ROC curves to assess whether the model 2 gives a significantly greater AUC compared with model 1, $P < 0.0001$
[b]DeLong's test for two correlated ROC curves to assess whether the model 3 gives a significantly greater AUC compared with model 2, $P = 0.342$

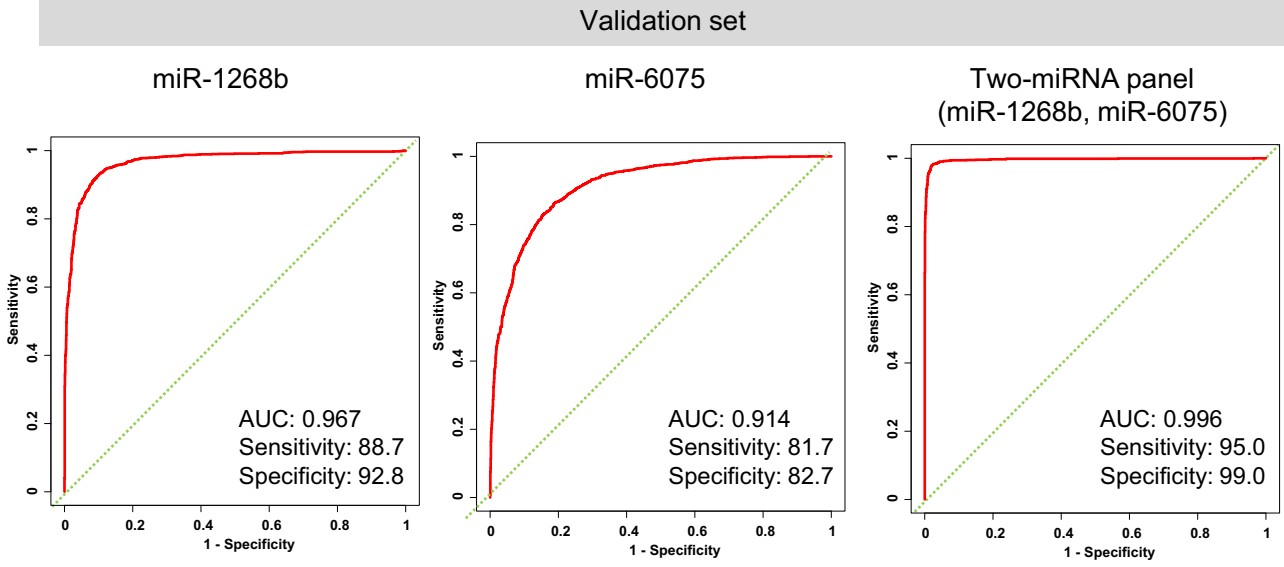

**Fig. 3 Development of a discrimination model to distinguish lung cancers from non-cancer participants. a** Receiver operating characteristic (ROC) curves for detecting lung cancer patients using miR-17-3p selected as the best single miRNA model in the discovery set. **b** Receiver operating characteristic (ROC) curves for detecting lung cancer patients using two miRNAs selected as best discrimination model in the discovery set. **c** Diagnostic index levels of miR-17-3p and two miRNAs selected as best discrimination model among lung cancer, non-cancer 1 and non-cancer 2. A diagnostic index score ≥ 0 indicated lung cancer and a diagnostic index score < 0 indicated the absence of lung cancer. model(miR-7-3p): (0.491213) × miR-17-3p−2.49845, model (miR-1268b): (−3.55955) × miR-1268b + 34.53362, model(miR-6075): (2.17625) × miR-6075−18.78233, model(miR-1268b + miR-6075): (−3.56049) × miR-1268b + (1.99039) × miR-6075 + 16.7999.

**Fig. 4 Diagnostic performance of the discrimination model in the validation set.** Receiver operating characteristic (ROC) curves for detecting lung cancer patients using two miRNAs selected as best discrimination model.

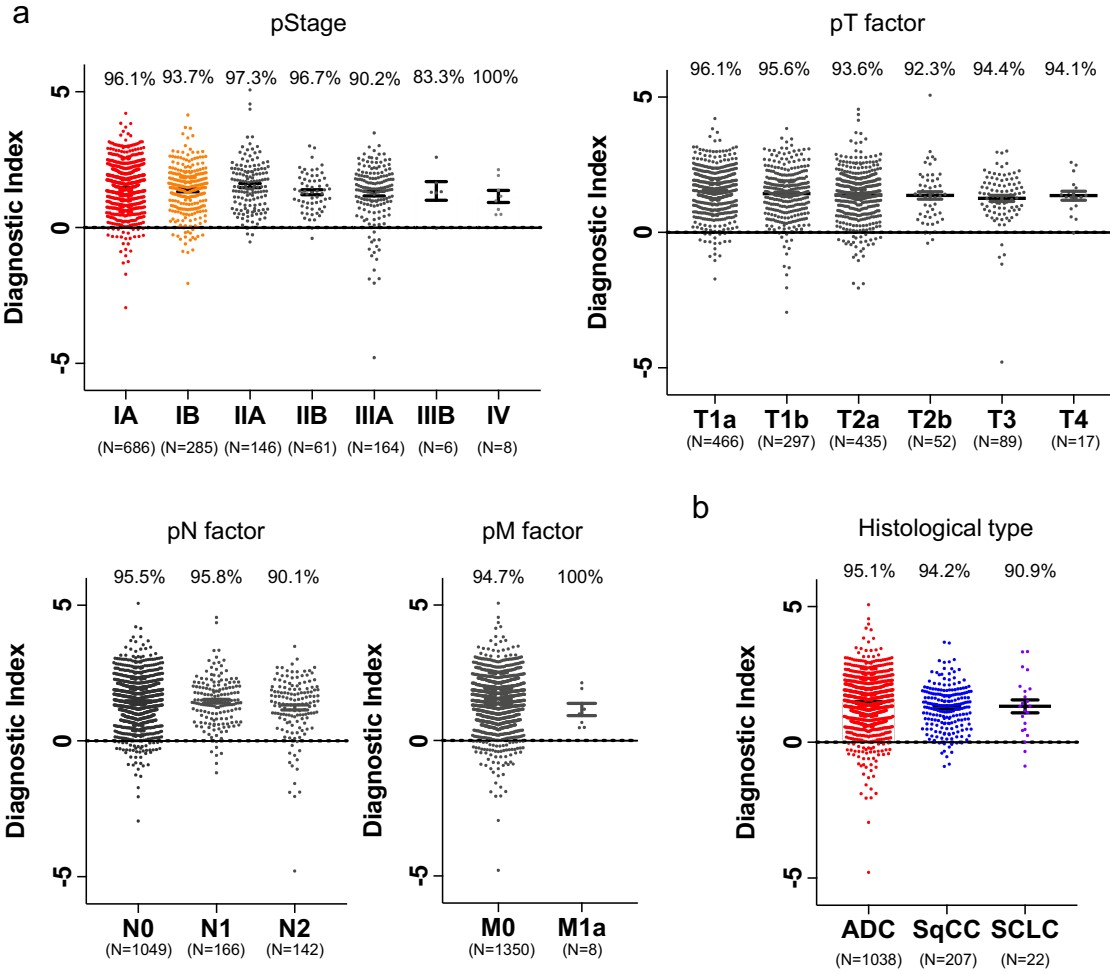

**Fig. 5 Diagnostic performance of the discrimination model at different stages and histological types of lung cancer. a** Diagnostic performance of the two selected miRNAs at different pathological TNM stages in the validation set. The diagnostic index showed high performance for all stages. Each positive rate is shown in the plot. **b** Diagnostic performance of the two selected miRNAs at different histological types in the validation set. The diagnostic index showed high performance for all histological types. Each positive rate is shown in the plot.

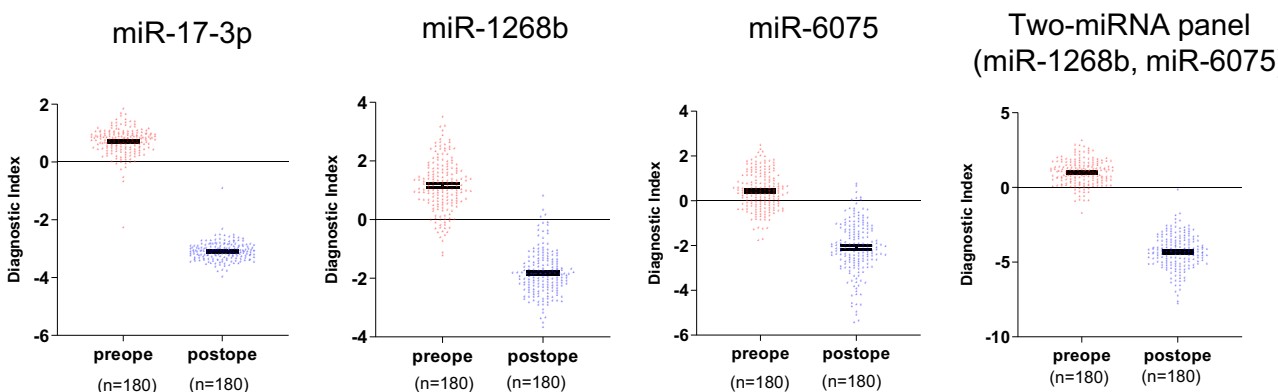

**Fig. 6 Comparison of diagnostic indexes between preoperative and postoperative serum samples from lung cancer patients.** Diagnostic index levels of miR-17-3p, miR-1268b, miR-6075, and two-miRNA panel (miR-1268b and miR-6075) were decreased after lung cancer resection. A diagnostic index score ≥ 0 indicated lung cancer and a diagnostic index score < 0 indicated the absence of lung cancer.

For example, serum miR-21, an oncogenic miRNA, is useful for detection lung cancer[20,21]. However, the results of previous studies are inconsistent, probably due to their small sample sizes and limited numbers of analyzed miRNAs. The largest previous study included 514 patients with lung cancer, but evaluated only one miRNA, miR-499[15]. Several studies were performed using miRNA panels with fewer than 34 miRNAs[14,22,23]. To address this issue, we conducted a national study to analyze comprehensive miRNA profiles in a large sample population. The diagnostic accuracy of lung cancer detection using our model (sensitivity and specificity, 99%) was much higher than those reported in previous studies. For example, Wang et al. performed

a meta-analysis of 28 articles involving 2121 non-small-cell lung cancer patients, and 1582 healthy controls, and reported that the overall pooled sensitivity and specificity of miRNA were 75 and 79%[24], respectively. One possible reason for the high diagnostic accuracy of our model is that we could select the best combinations of two miRNAs from the large number of pairwise combinations of 2588 miRNAs.

We identified the combination of miR-1268b and miR-6075 in serum as a biomarker for resectable lung cancer. To date, the function and serum expression profiles of these two miRNAs have not been extensively investigated. Zhu et al. reported that miR-1268b is upregulated in drug-sensitive breast cancer, and that ERBB2, a receptor tyrosine kinase, is a direct target gene of miR-1268b[25]. On the other hand, Kojima et al. reported that serum miR-6075 can detect pancreatic and biliary tract cancer with a sensitivity and specificity of 63.6 and 93.5%, respectively; their study analyzed 571 serum samples, including material from 100 patients with pancreatic or biliary tract cancer[26]. These reports suggest that elevated levels of the two miRNAs are associated with tumorigenesis and/or progression of lung cancer.

MiR-17-3p was the best single miRNA for detecting lung cancer in the discovery set (cross-validation score = 0.9087, Supplementary Table 1), although miR-1268b and miR-6075 combination achieved the highest accuracy as two miRNAs combination (cross-validation score = 0.9904, Supplementary Table 2). MiR-17-3p is a member of the miR-17-92 cluster, which has been reported to play oncogenic roles by promoting tumor cell proliferation[27,28], invasion[29], and suppressing apoptosis of tumor cells[30]. Previous reports have shown that miR-17-92 cluster were associated with the tumorigenesis in colorectal cancer[31,32], breast cancer[33], glioblastoma[34], skin cancer[35], gallbladder cancer[36], hepatocellular cancer[37], prostate cancer[29], B-cell lymphoma[30], and lung cancer[28]. Moreover, it has been reported that high expression level of serum miR-17-3p were correlated with poor prognosis in patients with colorectal cancer[38] and prostate cancer[39]. These reports regarding various types of cancers including lung cancer support our finding that elevated level of serum miR-17-3p highly correlated with lung cancer.

In this study, the serum levels of the two biomarker miRNAs (miR-1268b and miR-6075) were detected even in the early stages of lung cancer, and did not correlate with pathological stage. These characteristics are different from classical plasma tumor markers such as caricinoembrionic antigen. However, the diagnostic indexes of the selected biomarker miRNAs (miR-17-3p, miR-1268b, miR-6075 and the two-miRNA panel (miR-1268b and miR-6075) were dramatically decreased within 60 days after lung cancer resection. This finding suggests that these miRNAs are tumor-derived similar to plasma tumor markers. However, we have not identified the origins of these miRNAs. Therefore, the origins and detailed function of these miRNAs should be investigated in future studies.

Our results show that the miRNA profile of lung cancer patients is distinct from that of non-cancer participants, regardless of histological type and pathological TNM stage. Although adenocarcinoma is the most common histological type of lung cancer, other histological types account for approximately 20% of cases in the Japanese Lung Cancer Registry[40]. Therefore, accurate detection of lung cancers other than adenocarcinoma is also important. The 5-year overall survival rate of patients with stage I non-small-cell lung cancer is approximately 80%[2], whereas those of patients with stage III and IV non-small-cell lung cancer are 37 and 6%, respectively. Therefore, a high sensitivity (96%) for pathological stage IA non-small-cell lung cancer could dramatically decrease lung cancer-related mortality.

The National Lung Screening Trial revealed a 20% decrease in lung cancer mortality among high-risk participants who received

CT screening[3]. However, the most important limitation of screening by CT scan is a high false-positive rate, 64%, which results in unnecessary follow-up CT scans or invasive lung biopsies[5]. To address this issue, Sozzi et al. conducted a study to determine the diagnostic performance of a prespecified miRNA signature classifier algorithm in 939 participants retrospectively evaluated using samples prospectively collected within the randomized Multicenter Italian Lung Detection clinical trial comparing low-dose CT with usual care[41]. Their results showed that combining miRNA signature classifier and low-dose CT decreased the low-dose CT false-positive rate by 5-fold, from 19.4 to 3.7%[13]. Due to the high sensitivity and specificity of miRNA, first-line miRNA-based screening before low-dose CT could reduce the rate of unnecessary low-dose CTs and invasive biopsies in participants without lung cancer.

The absence of lung cancer in non-cancer controls was defined according to self-reported medical history and chest X-ray, and was not confirmed by CT scans or biopsies in most non-cancer participants from YMC. However, the National Lung Screening Trial reported that the incidence of lung cancer in high-risk populations is as low as 0.6%[3]. Therefore, it is reasonable to regard these participants as not having lung cancer. In addition, this study used retrospectively collected serum samples; therefore, storage conditions before microarray analysis were not strictly controlled, and this may have affected the results. Indeed, several studies reported that miRNAs are affected by various physical processes[42,43]. Because direct comparison between NCC Biobank samples and non-cancer control samples from YMC could introduce bias, we did not select biomarker miRNA candidates by comparing miRNAs between NCC lung cancer and YMC non-cancer samples. Rather, we used NCC non-cancer controls to choose miRNA candidates that were upregulated or downregulated in both NCC and YMC non-cancer samples relative to lung cancer samples. This process allowed us to exclude certain miRNAs showing alterations in serum levels only in NCC non-cancer patients, but not in YMC non-cancer participants. In addition, we recently started a prospective project to validate the general applicability of our findings in fresh serum samples; we will report the results in the near future.

In this study, the largest of its kind to date, analysis of serum miRNA levels from a large pool of participants with and without lung cancer identified a combination of two miRNAs, miR-1268b and miR-6075, as reliable markers for resectable lung cancer. The high sensitivity and specificity demonstrated by our results indicate the potential efficacy of this model in improving the early detection of resectable lung cancer, thereby reducing mortality and the use of unnecessary diagnostic procedures.

## Methods

**Lung cancer patients.** Serial serum samples were collected preoperatively from patients with lung cancer who underwent surgical resection at the National Cancer Center Hospital (NCCH). Serum collection was performed at outpatient department along with routine blood tests before surgery. These samples were registered in the NCC Biobank between 2008 and 2016 and stored at −20 °C until use. Clinical information for all samples was collected from the electronic medical charts of each patient. Exclusion criteria were as follows: past history of any other cancers before the collection of serum samples; preoperative treatment for lung cancer before the collection of serum samples; no information about smoking history; no information about pathological stage; interval between collection of serum samples and operation for lung cancer ≥180 days. Serum was also collected postoperatively from 180 lung cancer patients at the outpatient department within 60 postoperative days. Postoperative serum samples were also registered in the NCC Biobank and stored at −20 °C until use.

**Non-cancer participants.** Serum samples from study participants without cancer were collected at YMC (n = 1998) from patients at NCCH who were not diagnosed with cancer based on imaging or biopsy results (n = 207). Serum collection from non-cancer participants at NCCH were performed at outpatient department along with routine blood tests. Inclusion criteria for these study participants providing

non-cancer serum samples were no history of hospitalization during the last 3 months and no history of malignant disease. Information about smoking history was available for all samples. The serum samples were stored at −20 °C until use.

**Serum miRNA expression analysis**. Total RNA was extracted from aliquots (300 μL) of the serum samples with 3D-Gene® RNA extraction reagent (Toray Industries, Tokyo, Japan). Then, comprehensive miRNA profiling was performed with the 3D-Gene® miRNA Labeling kit and 3D-Gene® Human miRNA Oligo Chip (Toray Industries), which was designed to analyze 2588 miRNA sequences registered in miRBase release 21 (http://www.mirbase.org/)[19]. To ensure the quality of the microarray data, the following criteria for low-quality data were used: more than 10 flagged probes identified as "uneven spot images" by the 3D-Gene® Scanner and a coefficient of variation for negative control probes greater than 0.15. Samples fulfilling these criteria were excluded from further analysis. A miRNA was defined as present when the signal was greater than the mean + 2× the standard deviation of the negative control signal, from which the top and bottom 5%, as ranked by signal intensity, were removed. Once a miRNA was judged to be present, the average signal of the negative control was subtracted from the miRNA signal. Three internal control miRNAs (miR-4463, miR-2861, and miR-1493-p) were used to normalize the microarray signals as described previously[44–48]. The expression levels of these miRNAs in the present dataset were checked by geNorm log2 ranking analysis, and the results showed that these miRNAs were in the top 100 in serum samples from both NCCH and YMC (data shown in Supplementary Table 4), suggesting that these miRNAs are also suitable for normalization in this study. If the signal value was negative (or undetected) after normalization, the signal value was replaced with 0.1 on a base-2 logarithm scale.

**Statistics and reproducibility**. Samples were divided into the discovery and validation sets. To reduce bias in patient selection, one-to-one propensity score matching for age, sex, and smoking history was performed using the caliper-matching method in the validation set[49]. One-way analysis of variance for continuous variables and Chi-square tests for categorical variables were used to compare patient characteristics (age, sex, smoking history, histological type [World Health Organization classification, third edition][50], and pathological TNM stage [Union for International Cancer Control seventh edition][51]) in the discovery and validation sets.

Highly expressed miRNAs, with signal values greater than $2^6$, in more than 50% of the lung cancer or non-cancer samples in the discovery set were selected. In the discovery set, the best combinations of the identified miRNAs were explored using Fisher's linear discriminant analysis with leave-one-out cross-validation (Supplementary Methods). Briefly, we selected the best 20 discriminants by one miRNA, added one of residual miRNAs to make two-miRNA discriminants, and selected the best 20 discriminants by two miRNAs. Like this, we constructed 1- to 3-miRNA discriminants. Subsequently, we listed the best discriminants for each number of miRNAs, and finally selected the model of the best AUC with the least number of miRNAs using DeLong's test[52]. The solution of the discriminant (an 'index') ≥0 in the diagnostic model indicated the presence of lung cancer, whereas an index <0 indicated the absence of lung cancer. The performance of the diagnostic index was evaluated in the validation set (Fig. 2a).

Statistical analyses were conducted using R version 3.1.2 (R Foundation for Statistical Computing, http://www.R-project.org), MASS version 7.3–45, SPSS version 25 (IBM Corp., Armonk, NY), mutoss version 0.1–10, pROC version 1.8, compute.es version 0.2–4, hash version 2.26, and GraphPad Prism version 7 (GraphPad Software, La Jolla, CA). Unsupervised clustering and heat map generation using Pearson's correlation and Ward's method for linkage analysis as well as principal component analysis were performed using Partek Genomics Suite version 6.6. A two-sided p value less than 0.05 was defined as statistically significant.

**Ethical statement**. This study was approved by the Institutional Review Board of NCCH (#2015-376, #2016-249) and the Research Committee of Medical Corporation Shintokai YMC (#6019-18-3772). Written informed consent was obtained from each participant.

**Reporting summary**. Further information on research design is available in the Nature Research Reporting Summary linked to this article.

## Data availability

All miRNA microarray data have been deposited in the Gene Expression Omnibus (GEO) (https://www.ncbi.nlm.nih.gov/geo/) database (accession number: GSE137140). The authors declare that the data supporting the findings of this study are available within the paper and its supplementary information file. All source data underlying the graphs and charts are available within supplementary data files (Supplementary Data 1–6).

## Code availability

The algorithm for combinational optimization for multicandidate miRNAs and R scripts are available in Supplementary Information and Supplementary Data 7.

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

## Acknowledgements

The authors thank Tomomi Fukuda, Takumi Sonoda, Hiroko Tadokoro, Megumi Miyagi, Tatsuya Suzuki, Junpei Kawauchi, Makiko Ichikawa, and Kamakura Techno-Science Inc. for performing the microarray assays, Satoshi Kondo for technical support, Noriko Abe for the management of serum samples, Michiko Ohori for the management of personal information, Hitoshi Fujimiya for developing in-house analytic tools, and Kazuki Sudo for independent confirmation of participant eligibility. The National Cancer Center Biobank is supported by the National Cancer Center Research and Development Fund (29-A-1). This study was financially supported through a "Development of Diagnostic Technology for Detection of miRNA in Body Fluids" grant from the Japan Agency for Medical Research and Development (to TO).

## Author contributions

Conception and design. K.A., T.K., J.Matsuzaki, and T.O. Development of methodology. S.T., Y.A., E.N., J Miura, and T.O. Acquisition of data (acquired and managed patients, provided facilities, etc.). K.A., J.Matsuzaki, Y. Yoshida, K.N., H.S., K.K., and S.W. Analysis and interpretation of data (e.g., statistical analysis, biostatistics, computational analysis). K.K., T.K., Y. Yoshida, J.Matsuzaki and Y. Yamamoto. Writing, review, and/or revision of the manuscript. K.A., T.K., J.Matsuzaki, Y. Yamamoto, Y. Yoshida, K.N., S.T., Y.A., E.N., J Miura, H.S., K.K., S.W., and T.O. Administrative, technical, or material support (i.e., reporting or organizing data, constructing a database). Y. Yoshida, H.S., K.K., S.W., and T.O.

## Competing interests

S.T. is an employee of Toray Industries, Inc., the provider of the 3D-Gene® system. Y.A., E.N., and J Miura are employees of Dynacom Co., Ltd., the developer of the statistical script used for selecting the best miRNA combination. All other authors have no conflict of interest to declare.
