## [Peer Review File · Communications Biology]

Reviewers' comments:

Reviewer #1 (Remarks to the Author):

To develop a new screening method for lung cancer, the authors generated comprehensive miRNA profiles from 3744 serum samples obtained from 1566 patients with resectable lung cancer and 2178 participants with no disease. They created a diagnostic model by combining expression levels of two miRNAs, which showed a sensitivity of 99% and specificity of 99% in the validation set. The authors also found that their method exhibited high sensitivity regardless of histological type and pathological TNM stage.

The diagnostic accuracy with such high sensitivity and specificity of detecting lung cancer is remarkable. However, the usefulness of this new diagnostic technique requires additional evidence. I have several questions.

1) The authors used 208 lung cancer sample from NCC and 208 non-cancer samples from NCC. The serum test result may be affected by various conditions of sample collection, containers, process methods, etc. To minimize such bias, the authors excluded YMC samples at the time of developing the discovery set. However, other possibilities of bias still remain. Please provide the detail process of obtaining samples for lung cancer and non-cancer patients. For example, if the majority of samples were collected during the surgery for lung cancer patients, the microRNA profile can be affected dramatically. If the timing of sample collection was unique in lung cancer patients, the result might be related not to lung cancer but such condition. The authors should provide details of logistics for obtaining a sample in each group of patients.

2) Were there any patients in the non-cancer group, whose diagnostic index was positive and lung cancer was diagnosed later?

3) Had the author tested chronological samples from a lung cancer patient? If the authors had data showing positive diagnostic index preop and become negative after the surgery, then got positive with recurrence, it would have added additional confidence.

Reviewer #2 (Remarks to the Author):

The current manuscript entitled "Large-scale serum microRNA profiling for detection of resectable lung cancer" by Dr. Takahiro Ochiya group in National Cancer Center Research Institute, Japan described high accurate miRNA biomarkers in serum for detecting resectable lung cancer. Based on a large-scale serum cohort, the authors employed a high-throughput microRNA profiling followed by bio-statistical analyses and machine learning to discover and validate 1 to 3 miRNAs in combination for a high sensitive, specific and accurate detection for resectable lung cancer. In addition, this present study provides a clear logical platform for serum microRNA profiling for biomarker discovery, which can be also applied for other diseases. The written English is also pretty well.

Few comments are stated below:

1. The condition for collecting sera and/or preclinical protocol should be described in more details. Any difference in serum collection results in different sensitive or accurate?
2. The constant expression levels of the internal controls used in the current study should be demonstrated.
3. The analysis pipeline should be shown in a flow chart and included in a figure.
4. The origin and role of miR-17-3p, miR-1268b, miR-6075 or any other top ranking miRNAs in tumor progression, in particular the influence of tumor microenvironment, should be described and analyzed to support their de novo significance in tumorigenicity.

Reply to Reviewer #1

We would like to thank the reviewer for the constructive comments and suggestions regarding our manuscript. We modified the manuscript according to the comments and suggestions of the reviewer. We hope that the reviewer will find the new version satisfactorily re-revised and acceptable for publication in Communications Biology. Our point-by-point responses are listed below.

C1) To develop a new screening method for lung cancer, the authors generated comprehensive miRNA profiles from 3744 serum samples obtained from 1566 patients with resectable lung cancer and 2178 participants with no disease. They created a diagnostic model by combining expression levels of two miRNAs, which showed a sensitivity of 99% and specificity of 99% in the validation set. The authors also found that their method exhibited high sensitivity regardless of histological type and pathological TNM stage.

The diagnostic accuracy with such high sensitivity and specificity of detecting lung cancer is remarkable. However, the usefulness of this new diagnostic technique requires additional evidence. I have several questions.

The authors used 208 lung cancer sample from NCC and 208 non-cancer samples from NCC. The serum test result may be affected by various conditions of sample collection, containers, process methods, etc. To minimize such bias, the authors excluded YMC samples at the time of developing the discovery set. However, other possibilities of bias still remain. Please provide the detail process of obtaining samples for lung cancer and non-cancer patients. For example, if the majority of samples were collected during the surgery for lung cancer patients, the microRNA profile can be affected dramatically. If the timing of sample collection was unique in lung cancer patients, the result might be related not to lung cancer but such condition. The authors should provide details of logistics for obtaining a sample in each group of patients.

R1) Serum samples of lung cancer patients were collected at our outpatient department preoperatively. Serum samples of non-cancer participants from NCC were also collected at our outpatient department. Therefore there was no difference in methods of serum collection between lung cancer patients and non-cancer participants from NCC. We had corrected the manuscript as follows:

Page 5 line 13 to 17 in the Methods section:

Lung cancer patients

Serial serum samples were collected preoperatively from patients with lung cancer who underwent surgical resection at the National Cancer Center Hospital (NCCH). Serum collection was performed at outpatient department along with routine blood tests before surgery.

Page 6 line 4 to 8 in the Methods section:

Non-cancer participants

Serum samples from study participants without cancer were collected at Yokohama Minoru Clinic (YMC) (n = 1998) from patients at NCCH who were not diagnosed with cancer based on imaging or biopsy results (n = 207). Serum collection from non-cancer participants at NCCH was performed at outpatient department along with routine blood tests.

C2) Were there any patients in the non-cancer group, whose diagnostic index was positive and lung cancer was diagnosed later?

R2) Thank you for your suggestive comment. However, non-cancer participants from NCCH and YMC have not been followed-up after diagnosis of non-cancer. Therefore we cannot get additional information about late occurrence of lung cancer in this group.

C3) Had the author tested chronological samples from a lung cancer patient? If the authors had data showing positive diagnostic index preop and become negative after the surgery, then got positive with recurrence, it would have added additional confidence.

R3) Thank you for your invaluable suggestion. We agreed with your opinion that assessment over time will add additional confidence to our work. Therefore we performed comparable analysis of diagnostic index of three miRNAs (miR-17-3p, miR-1268b, miR-6075 and two miRNA panel (miR-1268b + miR-6075)) between preoperative and postoperative serum samples from 180 lung cancer patients. As a result, diagnostic indexes of three miRNAs (miR-17-3p, miR-1268b, miR-6075 and two miRNA panel (miR-1268b + miR-6075)) were dramatically decreased after operation. This finding suggests that three miRNAs were tumor-derived. Based on the results of additional experiments, we had corrected the manuscript as follows:

Page 5 line 23 to page 6 line 2 in the Methods section:

Serum was also collected postoperatively from 180 lung cancer patients at the outpatient department within 60 postoperative days. Postoperative serum samples were also registered in the National Cancer Center (NCC) Biobank and stored at -20°C until use.

Page 11 line 19 to page 12 line 2 in the Results section:

Comparison of the diagnostic indexes between preoperative and postoperative serum samples

Next, we compared the diagnostic indexes of three miRNAs between preoperative and postoperative serum samples from 180 lung cancer patients. The diagnostic indexes of miR-17-3p, miR-1268b, and miR-6075 and the two-miRNA panel (miR-1268 and miR-6075) were significantly decreased after surgery. (miR-17-3p, 0.71 ± 0.46 vs. -3.10 ± 0.32 , $p < 0.001$; miR-1268b, 1.15 ± 0.88 vs. -1.83 ± 0.75 , $p < 0.001$; miR-6075, 0.44 ± 0.85 vs. -2.10 ± 1.16 , $p < 0.001$; two-miRNA panel (miR-1268b and miR-6075), 0.98 ± 0.82 vs. -4.30 ± 1.15 , $p < 0.001$; Figure 6).

We added Figure 6 showing preoperative and postoperative diagnostic index of the selected miRNAs (miR-17-3p, miR-1268b, miR-6075 and two miRNA panel (miR-1268b and miR-6075)).

Page 14 line 3 to 11 in the Discussion section:

In this study, the serum levels of the two biomarker miRNAs (miR-1268b and miR-6075) were detected even in the early stages of lung cancer, and did not correlate with pathological stage. These characteristics are different from classical plasma tumor markers such as carcinoembryonic antigen (CEA). However, the diagnostic indexes of the selected biomarker miRNAs (miR-17-3p, miR-1268b, and miR-6075 and the two-miRNA panel (miR-1268 and miR-6075) were dramatically decreased within 60 days after lung cancer resection. This finding suggests that these miRNAs are tumor-derived similar to plasma tumor markers. However, we have not identified the origins of these miRNAs. Therefore, the detailed functions and origins of these miRNAs should be investigated in future studies.

Page 27 line 2 to 12 in the Figure legends:

Figure 6.

Comparison of diagnostic indexes between preoperative and postoperative serum samples from lung cancer patients

a. Diagnostic indexes of miR-17-3p in preoperative and postoperative serum from 180 lung

cancer patients.

b. Diagnostic indexes of miR-1268b in preoperative and postoperative serum from 180 lung cancer patients.

c. Diagnostic indexes of miR-6075 in preoperative and postoperative serum from 180 lung cancer patients.

d. Diagnostic indexes of two miRNA panel (miR-1268b and miR-6075) in preoperative and postoperative serum from 180 lung cancer patients.

Reply to Reviewer #2

C4) The current manuscript entitled “Large-scale serum microRNA profiling for detection of resectable lung cancer” by Dr. Takahiro Ochiya group in National Cancer Center Research Institute, Japan described high accurate miRNA biomarkers in serum for detecting resectable lung cancer. Based on a large-scale serum cohort, the authors employed a high-throughput microRNA profiling followed by bio-statistical analyses and machine learning to discover and validate 1 to 3 miRNAs in combination for a high sensitive, specific and accurate detection for resectable lung cancer. In addition, this present study provides a clear logical platform for serum microRNA profiling for biomarker discovery, which can be also applied for other diseases. The written English is also pretty well.

Few comments are stated below:

The condition for collecting sera and/or preclinical protocol should be described in more details. Any difference in serum collection results in different sensitive or accurate?

R4) Thank you for your constructive comment. Serum samples of lung cancer patients are collected at our outpatient department preoperatively. Serum samples of non-cancer participants from NCC were also collected at our outpatient department. Therefore there was no difference in methods of serum collection between lung cancer patients and non-cancer participants from NCC. We had corrected the manuscript as follows:

Page 5 line 13 to 17 in the Methods section:

Lung cancer patients

Serial serum samples were collected preoperatively from patients with lung cancer who underwent surgical resection at the National Cancer Center Hospital (NCCH). Serum

collection was performed at outpatient department along with routine blood tests before surgery.

Page 6 line 4 to 8 in the Methods section:

Non-cancer participants

Serum samples from study participants without cancer were collected at Yokohama Minoru Clinic (YMC) (n = 1998) from patients at NCCH who were not diagnosed with cancer based on imaging or biopsy results (n = 207). Serum collection from non-cancer participants at NCCH was performed at outpatient department along with routine blood tests.

C5) The constant expression levels of the internal controls used in the current study should be demonstrated.

R5) As you suggested, it is very important to show these internal controls are suitable in this lung cancer study, so we have checked these miRNA expression levels in the present dataset by geNorm log2 ranking analysis, the results showed that selected three miRNAs were listed in top 100 in serum samples from both NCCH and YMC (the data shown as Supplementary Table 4), suggesting that these miRNAs are also suitable for the normalization in lung cancer work. We had corrected the manuscript as follows:

Page 7 line 3 to 8 in the Methods section:

Three internal control miRNAs (miR-4463, miR-2861, and miR-1493-p) were used to normalize the microarray signals as described previously²⁰⁻²⁴. The expression levels of these miRNAs in the present dataset were checked by geNorm log2 ranking analysis, and the results showed that these miRNAs were in the top 100 in serum samples from both NCCH and YMC (data shown in Supplementary Table 4), suggesting that these miRNAs are also suitable for normalization in this study.

We added Supplementary Table 4 showing Top 100 miRNAs in NCCH and YMC serum samples.

C6) The analysis pipeline should be shown in a flow chart and included in a figure.

R6) Following your comments, we revised Figure 2a. In the original edition, Figure 2a shows

flow diagram of miRNA expression analysis only. In the revised version, Figure 2a shows entire work flow of the study including miRNA expression analysis and development of diagnostic model.

Page 25 line 11 to 14 in the Figure legends:

Figure 2.

Strategy for the selection of candidate miRNAs for lung cancer diagnosis.

a. Flow diagram of the (1) miRNA expression analysis and (2) development of the diagnostic model.

C7) The origin and role of miR-17-3p, miR-1268b, miR-6075 or any other top ranking miRNAs in tumor progression, in particular the influence of tumor microenvironment, should be described and analyzed to support their de novo significance in tumorigenicity.

R7) Thank you for your constructive comment. Following you and reviewer #1's comment, we performed additional experiment to compare diagnostic index of three miRNAs (miR-17-3p, miR-1268b, miR-6075 and two miRNA panel (miR-1268b and miR-6075)) between preoperative and postoperative serum samples from 180 lung cancer patients. As a result, diagnostic indexes of three miRNAs (miR-17-3p, miR-1268b, miR-6075 and two miRNA panel (miR-1268b + miR-6075)) were dramatically decreased after tumor resections. This finding suggests that three miRNAs were tumor-derived. Based on the results of this additional experiment, we had corrected the manuscript as follows:

Page 5 line 23 to page 6 line 2 in the Methods section:

Serum was also collected postoperatively from 180 lung cancer patients at the outpatient department within 60 postoperative days. Postoperative serum samples were also registered in the National Cancer Center (NCC) Biobank and stored at -20°C until use.

Page 11 line 19 to page 12 line 3 in the Results section:

Comparison of the diagnostic indexes between preoperative and postoperative serum samples

Next, we compared the diagnostic indexes of three miRNAs between preoperative and postoperative serum samples from 180 lung cancer patients. The diagnostic indexes of miR-17-3p, miR-1268b, and miR-6075 and the two-miRNA panel (miR-1268 and miR-6075) were significantly decreased after surgery. (miR-17-3p, 0.71 ± 0.46 vs. -3.10 ± 0.32 , $p <$

0.001; miR-1268b, 1.15 ± 0.88 vs. -1.83 ± 0.75 , $p < 0.001$; miR-6075, 0.44 ± 0.85 vs. -2.10 ± 1.16 , $p < 0.001$; two-miRNA panel (miR-1268b and miR-6075), 0.98 ± 0.82 vs. -4.30 ± 1.15 , $p < 0.001$; Figure 6).

We added Figure 6 showing preoperative and postoperative diagnostic index of the selected miRNAs (miR-17-3p, miR-1268b, miR-6075 and two miRNA panel (miR-1268b and miR-6075)).

Page 14 line 3 to 11 in the Discussion section:

In this study, the serum levels of the two biomarker miRNAs (miR-1268b and miR-6075) were detected even in the early stages of lung cancer, and did not correlate with pathological stage. These characteristics are different from classical plasma tumor markers such as carcinoembryonic antigen (CEA). However, the diagnostic indexes of the selected biomarker miRNAs (miR-17-3p, miR-1268b, and miR-6075 and the two-miRNA panel (miR-1268 and miR-6075) were dramatically decreased within 60 days after lung cancer resection. This finding suggests that these miRNAs are tumor-derived similar to plasma tumor markers. However, we have not identified the origins of these miRNAs. Therefore, the detailed functions and origins of these miRNAs should be investigated in future studies.

Page 27 line 2 to 12 in the Figure legends:

Figure 6.

Comparison of diagnostic indexes between preoperative and postoperative serum samples from lung cancer patients

- a. Diagnostic indexes of miR-17-3p in preoperative and postoperative serum from 180 lung cancer patients.
- b. Diagnostic indexes of miR-1268b in preoperative and postoperative serum from 180 lung cancer patients.
- c. Diagnostic indexes of miR-6075 in preoperative and postoperative serum from 180 lung cancer patients.
- d. Diagnostic indexes of two miRNA panel (miR-1268b and miR-6075) in preoperative and postoperative serum from 180 lung cancer patients.

R8) We added "Data availability" section in the Methods section according to your journal policies.

Page 8 line 22 to page 12 line 2 in the Methods section:

Data availability

All miRNA microarray data have been deposited in the Gene Expression Omnibus (GEO) (<https://www.ncbi.nlm.nih.gov/geo/>) database (accession number: GSE137140). The authors declare that the data supporting the findings of this study are available within the paper and its supplementary information file.

REVIEWERS' COMMENTS:

Reviewer #2 (Remarks to the Author):

This revised version has satisfied the raised comments. Thank you for the effort from the authors